# Lysine-Specific Demethylase 4D Is Critical for the Regulation of the Cell Cycle and Antioxidant Capacity in Goat Fibroblast Cells

**DOI:** 10.3390/biology12081095

**Published:** 2023-08-07

**Authors:** Xiaowei Chen, Yingnan Yang, Yu Cai, Hua Yang, Feng Wang, Mingtian Deng

**Affiliations:** 1College of Animal Science and Technology, Nanjing Agricultural University, Nanjing 210095, China; chenxiaowei@stu.njau.edu.cn (X.C.); 2020105030@stu.njau.edu.cn (Y.Y.); 2021205019@stu.njau.edu.cn (Y.C.); 2018205011@njau.edu.cn (H.Y.); 2Jiangsu Livestock Embryo Engineering Laboratory, College of Animal Science and Technology, Nanjing 210095, China

**Keywords:** antioxidation, DNA damage, ROS, Kdm4d, fibroblast

## Abstract

**Simple Summary:**

This paper focuses on the roles of lysine-specific demethylase 4D (Kdm4d) regarding its antioxidant capacity in goat fibroblast cells (GFCs). We identified 1013 down-regulated genes following the knockdown of Kdm4d in GFCs, and these genes were enriched in the cell cycle, DNA replication, mitotic processes, and the oxidative phosphorylation pathway. Consistently, we found the cell proliferation rate was significantly decreased after the knockdown of Kdm4d in the GFCs. Of note, the mRNA and protein expression levels of superoxide dismutase 2 (SOD2), one of the major antioxidant enzymes, was decreased, while the reactive oxygen species (ROS) level was significantly increased in the Kdm4d knockdown GFCs. The expression of γH2A histone family member X (γH2AX) was increased significantly, indicating the presence of DNA double-strand breaks after the knockdown of the Kdm4d enzyme. Taken together, Kdm4d is essential for DNA replication, the cell cycle, and antioxidant capacity. The knockdown of Kdm4d repressed the expression of SOD2 and increased the generation of ROS, which led to DNA damage in GFCs. Our data will be helpful for understanding the mechanism underlying antioxidant capacity regulation in fibroblast cells.

**Abstract:**

Oxidative damage to skin fibroblast cells is a causative factor in many skin diseases. Previous studies have reported that lysine-specific demethylase 4D (Kdm4d) is involved in DNA replication, but its role on antioxidant capacity remains unclear. In the present study, we used goat fibroblast cells (GFCs) as the research model and identified 504 up-regulated and 1013 down-regulated genes following the knockdown of Kdm4d, respectively. The down-regulated genes of this enzyme were found to be enriched in the cell cycle, DNA replication, mitotic processes, and the oxidative phosphorylation pathway, as previously revealed from gene ontology (GO), Kyoto encyclopedia of genes and genomes (KEGG), and gene set enrichment analysis (GSEA), suggesting vital roles of the Kdm4d enzyme in the cell cycle and in antioxidant regulation. To this end, we found the cell proliferation rate was significantly decreased after the knockdown of Kdm4d. Moreover, both the mRNA and protein expression levels of superoxide dismutase 2 (SOD2), one of the major antioxidant enzymes, was decreased, while the reactive oxygen species (ROS) level was significantly increased in Kdm4d knocked-down cells. In addition, the expression of γH2A histone family member X (γH2AX) increased significantly, indicating the presence of DNA double-strand breaks after the knockdown of the Kdm4d enzyme. In conclusion, the knockdown of Kdm4d inhibited DNA replication and the cell cycle, repressed the expression of SOD2, and increased the generation of ROS, which led to the production of DNA damage in GFCs. Our data will be helpful for understanding the mechanism underlying antioxidant capacity regulation in fibroblast cells.

## 1. Introduction

The incidence of skin cancer has been increasing and remains highly prevalent with a lack of effective treatment strategies, making it a global public health concern [1]. The etiology of skin cancer is complex; genetic or environmental factors lead to erroneous DNA repair mechanisms, ultimately triggering skin cancer [2,3]. For example, prolonged exposure to ultraviolet radiation B would directly interact with DNA molecules and produce pyrimidine dimers, which hinder transcription and replication and disrupt the integrity of the DNA double helix, causing DNA lesions; ultraviolet radiation A generates reactive oxygen species (ROS) that inflict secondary damage to DNA [4]. Fibroblast cells serve as one of the important repair cells and components of the skin [5], and show a strong ability to divide and proliferate, which makes them an excellent model for exploring various treatment strategies of skin cancer [6].

Oxygen-derived free radicals and non-radical reactive oxygen species, which are mainly generated through the mitochondria and NADPH oxidases [7,8], play essential roles in cell signaling and homeostasis [9]. Wnt signaling induces replicative senescence in mouse embryonic fibroblast cells via increased mitochondrial biogenesis along with the consequent increase in ROS-mediated damage [10]. These, and subsequent studies, have demonstrated that ROS play a crucial role in the regulation of gene activation, cell growth, and intracellular chemical reactions by either acting as signaling molecules directly or mediating redox reactions [11]. However, high levels of ROS lead to oxygen toxicity and oxidative stress. Previous studies have revealed that excessive ROS severely damages cell structures and cell metabolism by causing the dysfunction of proteins and DNA structures [12]. The presence of excessive ROS in the cells also lead to skin-related diseases [13,14]. Therefore, it is critical to maintain moderate concentration of ROS in the cellular system.

In mammals, the antioxidant system inhibits the chain reaction of ROS. Disruption of the oxidative–antioxidative balance leads to the decrease in biological system detoxification activity and the ability to repair damage [15]. A previous study has documented that superoxide dismutase 2 (SOD2), catalase (CAT), and glutathione peroxidase (GPX) are major antioxidant enzymes. It was reported that SOD2 promptly reduced ROS levels by breaking down excess superoxide ions into oxygen and hydrogen peroxide in mitochondria [16]. CAT functions to break down hydrogen peroxide, the catalytic product of SOD2, into water and oxygen [17], respectively, while the role of GPX is reducing toxic peroxides into hydroxyl compounds. Loss- and gain-function studies have revealed critical roles of SOD2 and GPX4 in the protection of oxidative stress-induced damage [18,19]. These, and subsequent studies, have suggested that the oxidative–antioxidant balance is the key to maintaining healthy biological systems [20].

Epigenetic modifications play pivotal roles in enhancing the self-repair ability of cells and in delaying cell aging [21], which may exhibit a certain effect on the antioxidant function of cells. The tri-methylation of H3K9 (H3K9me3) has been linked to the silencing of heterochromatin [22]. Lysine-specific demethylase 4D (Kdm4d), encoding 523 amino acids, is a histone demethylase recruited to chromatin and catalyzes the removal of the di-methylation of H3K9 (H3K9me2) and H3K9me3 [23]. In addition, Kdm4d is recruited to DNA damage sites through the RNA-binding domains at the N-terminal and C-terminal ends and repair the broken duplexes [23,24]. The knockout of Kdm4d blocked the recruitment of the cell division-related gene Cdc45 and the cell proliferation-related gene PCNA, resulting in DNA replication defects [25], suggesting that Kdm4d is critical for proliferation.

In the present study, we aimed to investigate the role of Kdm4d in the regulation of the cell cycle and the antioxidant capacity in goat fibroblast cells. We reported that *Kdm4d* interference inhibited DNA replication and the cell cycle and was involved in the cellular oxidative phosphorylation process and repressed the expression of SOD2, increasing the level of ROS generation, which led to the accumulation of DNA damage in GFCs. These data provided novel insights into the repair of oxidative cellular damage in fibroblast cells.

## 2. Materials and Methods

All the experimental procedures involving animals were following the National Research Council’s publication titled the *Guide for the Care and Use of Laboratory Animals* and was approved by the Institutional Animal Care and Use Committees at Nanjing Agricultural University (NJAU-LAWER-2021037).

### 2.1. Sample Collection and Fibroblast Cells Culture

The goat fibroblast cells (GFCs) used in this study were cryopreserved in the Jiangsu Livestock Embryo Engineering Laboratory and were cultured as described previously [26]. Briefly, fibroblast cells were isolated from the ear of the transgenic cloned goats (aged 60 days old). Ear tissues were minced and cultured in DMEM (Gibco, CA, USA) supplemented with 15% (*v*/*v*) fetal bovine serum (FBS [Gibco, CA, USA]) in 35 mm tissue culture plates at 37 °C with 5% CO_2_ and saturated humidity. After 10 days in the culture, the explants were removed and the cells were harvested, passaged, and then expanded. The cells were harvested or passaged into the next generation when they were 90% confluent. 

### 2.2. Localization of Kdm4d

Since no commercial antibody of Kdm4d was available in goat, we constructed a fusion expression vector of Kdm4d, as described in our previous study [27]. Briefly, total RNA was extracted from the goat testis using Trizol reagent and was reversed transcribed into cDNA using the Prime Script reagent kit with gDNA Eraser (6210A, TaKaRa, Dalian, China), following the manufacturer’s recommendation. The goat Kdm4d coding sequence was amplified from the goat testis cDNA library using the phanta max super-fidelity DNA polymerase (#P505-d1, Vazyme, Nanjing, China). PCR products were cloned into the pcDNA3-EGFP plasmid (Addgene plasmid #13032) using the NEBuilder HiFi DNA assembly master mix (E2621S, NEB, Beijing, China). To confirm the sequence of the fusion expression vector of Kdm4d, we performed restriction endonuclease digestion using BamHI and EcoRI, and the target fragment size was determined using gel electrophoresis and sanger sequencing. The primers used in the amplification of the Kdm4d coding sequence are listed in Table 1.

### 2.3. Cell Transfection

Small interfering RNAs (siRNAs) were formulated using the BLOCK-iT™ RNAi Designer tool and were synthesized at GeneParma (Shanghai, China). GFCs at passage 7 were cultured in 12-well plates and were then transfected with either the fusion expression vector or with the siRNA of Kdm4d when the cells were 60% confluent using the Lipofectamine 3000 reagent (L3000001, Thermo Fisher Scientific, Waltham, MA, USA), according to the manufacturer’s instructions. GFCs were either collected at 48 h for gene expression analysis or at 72 h for protein expression analysis using immunofluorescence staining and Western blotting, respectively. The sequences of siRNA against *Kdm4d* have been listed in Table 2.

### 2.4. RNA Sequencing

RNA-seq and differential gene expression (DEG) was conducted following a previously published study [28]. Briefly, genomic DNA was removed using DNase I. The purified mRNA was then reverse transcribed into cDNA, followed by second-strand cDNA synthesis. After the cDNA was fragmented and purified, paired-end 150 bp sequencing was performed on HiSeq X Ten by Annoroad Gene Technology Company. Three mRNA libraries were constructed each for the NC (negative control) and Kdm4d knockdown groups. All sequenced reads were trimmed to remove adaptors and low-quality bases using fastp (version 0.19.6). All the reads that passed through quality control were mapped to the goat reference genome (ARS1, NCBI) using hisat2 (version 2.2.1) software with default settings. Uniquely mapped reads were subsequently assembled into transcripts guided with the reference annotation using featureCounts. Gene expression levels were quantified by calculating normalized fragments per kilobase of exon per million mapped fragments (FPKM) values. Differential expression analysis was performed using DESeq2 (version 3.11). Genes with log_2_ (fold change) > 1 or log_2_ (fold change) < −1 and with statistical significance (*p*-value < 0.05) were deemed as differential expression genes (DEGs). 

Goat genes were transformed into homologous human genes using the biomaRt R package (version 2.40.5). Gene ontology (GO) and Kyoto encyclopedia of genes and genomes (KEGG) pathway enrichment analyses of up-regulated and down-regulated DEGs were conducted separately using the clusterProfiler R package (version 3.12.0). GO and KEGG terms with an FDR adjust *p*-value < 0.05 were deemed as statistically significant. Gene set enrichment analysis (GSEA) was performed using the GSEA software and additional resources within the software. Briefly, a tab-delimited file that contains DEGs was generated and used as an expression dataset. The additional resources c2.cp.kegg.v7.1.symbols.gmt and c5.all.v7.1.symbols.gmt were selected as a gene sets database. Other parameters were used as default.

### 2.5. Gene Expression Analysis

Gene expression analysis were performed as described in our previous study [29]. Briefly, total RNA was extracted from GFCs using Trizol reagent. Reverse transcription was performed using the TransScript II Reverse Transcription Kit (AH101-02, TransGen, Beijing, China), following the manufacturer’s suggestions. Quantitative real-time PCR (qPCR—reaction volume, 20 µL) was performed on an ABI 7300 real-time PCR system (A28108, Applied Biosystems, Waltham, MA, USA). The PCR reaction consisted of 10 µL of SYBR Green PCR Master Mix, 10 µM of forward and reverse primers, and 1.0 µL template cDNA, in a total volume of 20 µL. The PCR conditions were as follows: initial denaturation at 94 °C for 30 s, followed by 40 cycles at 94 °C for 5 s, 60 °C for 30 s, annealing at 95 °C for 15 s, 60 °C for 30 s, and extension at 95 °C for 1 s. A melting curve was performed to confirm the identity of each PCR product. Gene expression was normalized to ACTB, a housekeeping gene. The relative amount of each transcript present in each cDNA sample was calculated using the 2^−ΔΔCT^ method. The primers that were used in qPCR are shown in Table 1. GraphPad Prism 8 was used to perform the one-way ANOVA statistical test to assess the results.

### 2.6. Immunofluorescence Staining

Immunofluorescent staining was performed as previously described [30]. Briefly, GFCs were fixed with 4% paraformaldehyde (PFA) in PBS for 15 min at room temperature (RT), followed by permeabilizing with 0.25% Triton X-100 for 30 min at RT. After blocking in 5% BSA in PBS for 2 h at RT, GFCs were incubated with the corresponding primary antibody overnight at 4 °C. The GFCs were then washed in blocking solution and incubated for 1 h at RT with Alexa Flour 488-conjugated anti-mouse IgG secondary antibodies (ab150117, 1:200 dilution, Abcam, Cambridge, UK) and/or Alexa Fluor 594 conjugate anti-rabbit IgG (H&L) secondary antibodies (8889, 1:250 dilution, Cell Signaling Technology, Danvers, MA, USA). The nuclei were stained with 1 mg/mL of 4′, 6-diamidino-2-phenylindole (DAPI) for 5 min at RT and the GFCs were then placed on a glass slide and observed with a LSM710 laser scanning confocal microscope (Carl Zeiss, Oberkochen, Germany) under the same condition. The antibodies used in immunofluorescent staining are shown in Table 3. 

### 2.7. Western Blot Analysis

The Western blotting assay was performed as previously described [27]. Briefly, proteins of GFCs were extracted using RIPA protein lysis solution containing 1% protease inhibitor. After denaturation, 10 µg of protein samples were loaded and separated through electrophoresis and transferred onto PVDF membranes using a protein rapid transfer apparatus. After blocking in 5% BSA for 2 h at RT, PVDF membranes were incubated with the corresponding primary antibody overnight at 4 °C, then washed in TBST solution, and incubated with HRP-goat anti-rabbit/mouse IgG for 1 h at RT. Subsequently, immunoblotting was visualized using enhanced ECL ultra-sensitive luminescence fluid (32209, Thermo Fisher, Waltham, MA, USA) and exposed with Image Quant LAS 400 (Fiji film, Tokyo, Japan). Fold changes of protein levels were analyzed using Image J software (Wayne Rasband, Bethesda, MD, USA) and normalized to ACTB. The antibodies that were used for the Western blotting assay are shown in Table 3.

### 2.8. Cell Proliferation Assay

The cell proliferation assay was performed using the Cell Proliferation EdU Image Kit (C0085S, Beyotime, Shanghai, China), following the manufacturer’s instructions. Briefly, transfected GFCs were spread on a round coverslip and incubated with 100 µM EdU. The GFCs were then fixed with 4% PFA for 15 min at RT, and permeabilized using 0.25% Triton X-100 for 30 min at RT. Subsequently, GFCs were incubated with the reaction solution for 30 min. After washing with 1× PBS, cell nuclei were stained with Hoechst (Abcam, 1:2000). The coverslip was then sealed with antifade mounting medium and photographed under the same conditions.

### 2.9. Cellular Reactive Oxygen Detection

Cellular reactive oxygen was detected using the Reactive Oxygen Species Assay Kit (S0033S, Beyotime, Shanghai, China) according to the manufacturer’s protocol. DCFH-DA was diluted with serum-free culture medium at 1:1000 to a final concentration of 10 µM. The GFCs were incubated with the diluted DCFH-DA for 20 min at 37 °C and observed directly under a laser microscope. The fluorescence intensity of each cell was measured using Image J.

### 2.10. Cellular Mitochondrial Membrane Potential Detection

Cellular mitochondrial membrane potential was detected using the Mitochondrial Membrane Potential Assay Kit with JC-1 (C2003S, Beyotime, Shanghai, China), according to the manufacturer’s protocol. JC-1 (200×) was diluted with ultrapure water and mixed with JC-1 buffer to form the JC-1 working buffer. The GFCs were incubated with the JC-1 working buffer for 20 min at 37 °C and observed directly under a laser microscope. The fluorescence intensity of each cell was measured using Image J.

## 3. Results

### 3.1. Cellular Localization of Kdm4d in Goat Fibroblast Cells

The protein sequence of goat Kdm4d exhibited the highest homology with sheep, reaching a level of 98.2%, and 94.4% with cattle, while the homology with humans and mice was lower, reaching levels of 67.2% and 62.8%, respectively (Figure 1A). Since no commercial antibody was available for Kdm4d in goat, we successfully constructed a fusion overexpression vector for the cellular localization of Kdm4d (Figure 1B). The EGFP signal was observed in the nuclei of GFCs (Figure 1C,D), indicating that Kdm4d was mainly expressed in the nuclei of GFCs.

### 3.2. Increased H3K9 Methylation in Kdm4d Knockdown Fibroblast Cells

We next performed the knockdown of Kdm4d. As shown in Figure 2A, the expression of Kdm4d was decreased in Kdm4d knockdown GFCs, as revealed with qPCR (*p* < 0.01). The levels of H3K9me2 and H3K9me3 were elevated (*p* < 0.05, Figure 2B–D), while the signal intensity of H3K9me1 was not significantly different in Kdm4d knockdown GFCs compared to the controls. These data suggest that Kdm4d was successfully knocked down, which led to the observed higher level of H3K9me2/3 in GFCs.

### 3.3. Transcription Profile of Kdm4d Knockdown GFCs

To explore the roles of Kdm4d in GFCs, we performed RNA-seq in Kdm4d knockdown GFCs. Principal component analysis (PCA) and clusters analysis showed that the cells were clearly clustered into two groups (Figure 3A,B). Additionally, we identified 504 up-regulated and 1013 down-regulated genes in Kdm4d knockdown GFCs, respectively (Figure 3C). The heatmap displayed the well-defined clustering of these differentially expressed genes and the expression changes between the NC group and the Kdm4d interference group (Figure 3D). GO enrichment analysis showed that the down-regulated differential genes were mainly enriched in organelle fission, nuclear division, chromosome segregation, positive regulation of the cell cycle, DNA replication, and mitotic processes (Figure 3E). KEGG enrichment analysis revealed that the down-regulated DEGs were mainly involved in signaling pathways related to cellular senescence, lipid and atherosclerosis, the cell cycle, the FoxO signaling pathway, the thyroid hormone signaling pathway, cancer-related pathways, and DNA replication (Figure 3F). To confirm these results, we also performed GSEA analysis. As shown in Figure 4A–F, these genes were related to the regulation of DNA replication, cellular response to the DNA damage stimulus, chromosomes, ATP-dependent activity, acting on DNA, DNA repair, the mitotic cell cycle, and in the regulation of the cell cycle, suggesting that Kdm4d is involved in the repair of DNA damage and in cell cycle regulation.

### 3.4. Kdm4d Regulates the Cell Cycle of Goat Fibroblast Cells

DNA replication is one of the important parts of the cell cycle. We next studied the role of Kdm4d in the cell cycle. As shown in Figure 5A, the expression of the DNA repair-related gene DNA2, DNA genetic stability-related gene *FEN1*, DNA ligase-related gene *LIG1*, DNA replication-initiation-related gene *MCM4*, DNA replication-associated gene *POLD1*, and cell proliferation-associated gene *PCNA* was decreased in Kdm4d knockdown GFCs (*p* < 0.05). Moreover, the cell proliferation rate showed a significant decrease in Kdm4d knockdown GFCs compared to the NC group (*p* < 0.05, Figure 5B), as revealed from the EdU cell proliferation assay. To further investigate whether Kdm4d directly affects the cell cycle, we examined the expression of cell cycle-related genes. As shown in Figure 5A, the expression of *CCND1*, *CDK1*, *CDK2*, *CDK4,* and *UHRF1* were significantly decreased after the knockdown of Kdm4d (*p* < 0.01). The protein expression levels of UHRF1 and CCND1 were significantly decreased in the Kdm4d knockdown group (*p* < 0.05, Figure 5C and Appendix A), suggesting that Kdm4d is important in regulating the cell cycle process of GFCs. Our data indicate that interfering with Kdm4d is essential for the proliferation of GFCs.

### 3.5. Kdm4d Regulates the Antioxidant Capacity of Goat Fibroblast Cells

GSEA analysis also drew our attention to the oxidative phosphorylation pathway (Figure 6A), indicating that Kdm4d might play an important role in regulating the oxidative phosphorylation pathway. As shown in Figure 6B, the expression of the mitochondrial activity and energy metabolism-related genes *ND6*, *NDUFA1*, *NDUFA13*, *NDUFB3*, and *NDUFB9* were significantly increased in the Kdm4d knockdown GFCs (*p* < 0.01). Moreover, the cellular mitochondrial membrane potential detection illustrated a significant increase in the membrane potential of mitochondria in the Kdm4d knockdown GFCs (*p* < 0.05, Figure 6C), which indicates that Kdm4d may play a certain active role on mitochondrial activity through the oxidative phosphorylation pathway.

Since ROS is one of the by-products of oxidative phosphorylation and is closely related to cellular antioxidant capacity, to this end, we detected the levels of cellular reactive oxygen in Kdm4d knockdown cells. As shown in Figure 6D, the reactive oxygen level of GFCs showed a significant increase after interfering with Kdm4d. Moreover, the expression of the antioxidant-related genes *SOD2*, *GPX1*, *GPX3*, and *GPX8* were significantly decreased in Kdm4d knockdown GFCs (*p* < 0.05, Figure 6E). Meanwhile, the protein expression of SOD2 was significantly decreased in the Kdm4d-interfered GFCs (*p* < 0.05, Figure 6F), while the protein expression of CAT was not statistically significant, consistent with the mRNA expression. In addition, the protein H2AX was activated to form γH2AX, indicating the presence of DNA double-strand breaks, and the expression of γH2AX was significantly increased in the Kdm4d knockdown GFCs (*p* < 0.05, Figure 6F and Appendix A). These data indicate that Kdm4d is essential for the antioxidant capacity of GFCs.

## 4. Discussion

DNA damage is one of the results of cellular oxidative damage. Accumulating evidence has demonstrated that Kdm4d is pivotal for cell proliferation and DNA damage repair by being recruited to DNA damage sites under the actions of the polymerase PARP1 and the DNA repair proteins Rad51 and p53 [23,24]. Nevertheless, the molecular mechanism and its role in the antioxidation of GFCs have not yet been elucidated. Herein, the present study revealed that the knockdown of Kdm4d inhibited DNA replication and the cell cycle and repressed the expression of SOD2 to reduce the generation of ROS, which typically lead to DNA damage in GFCs.

Kdm4d catalyzes the removal of H3K9me2/3. As there is no commercial antibody available for Kdm4d in goat, we found the expression of Kdm4d in the nuclei of GFCs using a fusion overexpression vector. qRT-PCR results showed that Kdm4d mRNA was significantly reduced in the GFCs, while the levels of H3K9me2 and H3K9me3 were increased in the Kdm4d knockdown GFCs. Consistently, a previous study indicated that Kdm4d is able to reduce the negative effects of histone H3K9 methylation as a histone demethylase [23].

H3K9me2/3 is associated with gene repression; establishment of H3K9me2/3 through Kdm4d knockdown may lead to abnormal gene expression. To this end, we identified 504 up-regulated and 1013 down-regulated genes after the knockdown of Kdm4d, respectively. It is noteworthy that these down-regulated genes were mainly enriched in the cell cycle and DNA replication pathways. The down-regulation of the DNA replication-related factors *DNA2*, *FEN1*, *LIG1,* and the proliferate-related factor *PCNA* suggest the blocking of DNA replication and proliferation, as revealed from the EdU assay. In line with our data, a previous study showed that Kdm4d is recruited to DNA damage sites through the RNA-binding domains at the N-terminal and C-terminal ends and repairs the broken duplexes [23,24]. Moreover, Kdm4d facilitates the formation of the pre-initiative complex by recruiting Cdc45, PCNA, and polymerase delta to regulate DNA replication [25]. Our data, and these studies, demonstrate that Kdm4d is critical for DNA replication in mammalian cells.

Kdm4d may participate in the cell cycle to indirectly regulate cell proliferation; our results uncovered this noted effect of Kdm4d on the cell cycle. Positive cell cycle regulators, such as *CDK1*, *CDK4*, *CCND1*, and *UHRF1*, were transcriptionally inhibited after Kdm4d interference in goat GFCs. In addition, the protein expression of CCND1 and PCNA also confirmed the above results. Previously, there was evidence which demonstrated that Kdm4d could regulate adult cardiomyocyte cycling and proliferation by activating different cell cycle regulators [31], consistent with our present results. These findings, and our results, indicate that the dysregulation of Kdm4d affects cell proliferation and the cell cycle.

The antioxidant system is very important for the damage repair of protein and the DNA structure [32]. It is critical to maintain the ROS level in cell systems. In the present study, we screened the oxidative phosphorylation pathway through GSEA analysis, which suggested that Kdm4d may be involved in the antioxidant functions of GFCs. The expression of the mitochondria function and oxidative phosphorylation markers *ND6*, *NDUFA1*, *NDUFB3,* and *NDUFS3* were significantly upregulated in the Kdm4d knockdown GFCs. Mitochondria converts nutrients into adenosine triphosphate (ATP) through oxidative phosphorylation and plays an important role in biological processes, such as cell death, senescence, DNA replication, and autophagy [33,34]. It has been shown that mitochondria are involved in regulating the production and sequestration of reactive oxygen species through the oxidative phosphorylation pathway [35], congruent to our reactive oxygen assay results.

SOD2 is mainly localized in the mitochondria and promptly reduces ROS levels by breaking down excess superoxide ions into oxygen and hydrogen peroxide [36], respectively, while the role of GPX is to reduce toxic peroxides into hydroxyl compounds, as the excellent indicators of cellular antioxidant capacity. As anticipated, *SOD2*, *GPX1*, *GPX3,* and *GPX8* were all significantly downregulated in the Kdm4d interference groups. The protein expression of SOD2 also confirmed the above results. CAT is responsible for breaking down hydrogen peroxide, the catalytic product of SOD2, into water and oxygen, respectively [17]. These data suggest that Kdm4d is able to directly remove reactive oxygen species through the actions of SOD2. Herein, it is possible that Kdm4d plays an important role in the catabolism of superoxide ions and in the process of converting peroxides into hydroxyl compounds, thereby inhibiting the production of ROS and increasing the antioxidant capacity of GFCs. Of note, the elevated expression of the protein γH2AX also indicated DNA double-strand breaks, which was induced by an excess level of intracellular ROS in Kdm4d knockdown GFCs, consistent with previous findings [36,37].

## 5. Conclusions

In conclusion, our results revealed that the knockdown of Kdm4d inhibited DNA replication and the cell cycle of goat fibroblast cells. Moreover, Kdm4d knockdown repressed the expression of SOD2 to reduce the generation of ROS, which may lead to DNA damage, thereby playing an essential role in enhancing the antioxidant capacity of goat fibroblast cells. These findings contribute to a better understanding of the molecular mechanism of Kdm4d in regulating the antioxidant capacity of goat fibroblast cells and provide new insights for the gene therapy of oxidative damage.

## Figures and Tables

**Figure 1 biology-12-01095-f001:**
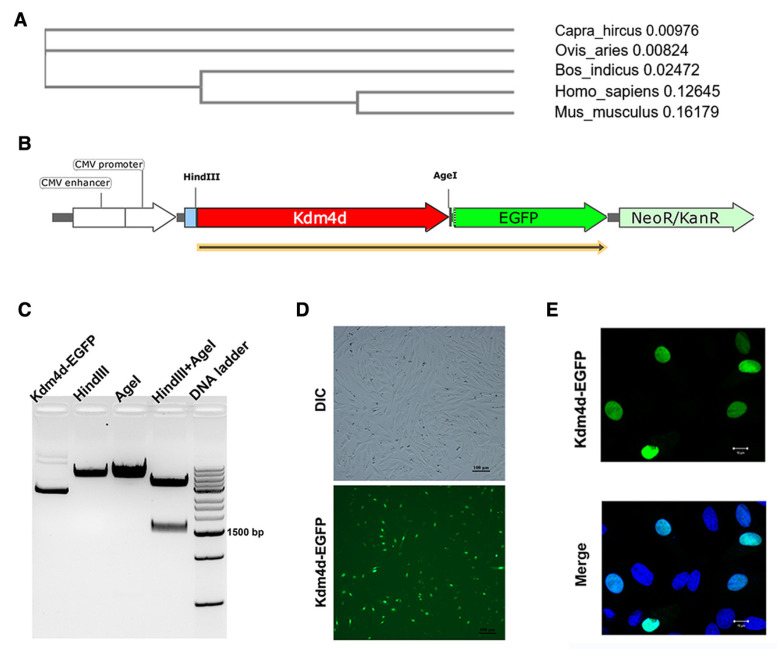
Cellular localization of Kdm4d in goat fibroblast cells. (**A**) Evolutionary tree generated from Kdm4d protein sequences from different species; (**B**) schematic diagram of overexpression fusion vector construction; (**C**) the product after constructing the overexpression vector using gel electrophoresis; and (**D**,**E**) the cellular localization of Kdm4d in goat fibroblast cells using immunofluorescence.

**Figure 2 biology-12-01095-f002:**
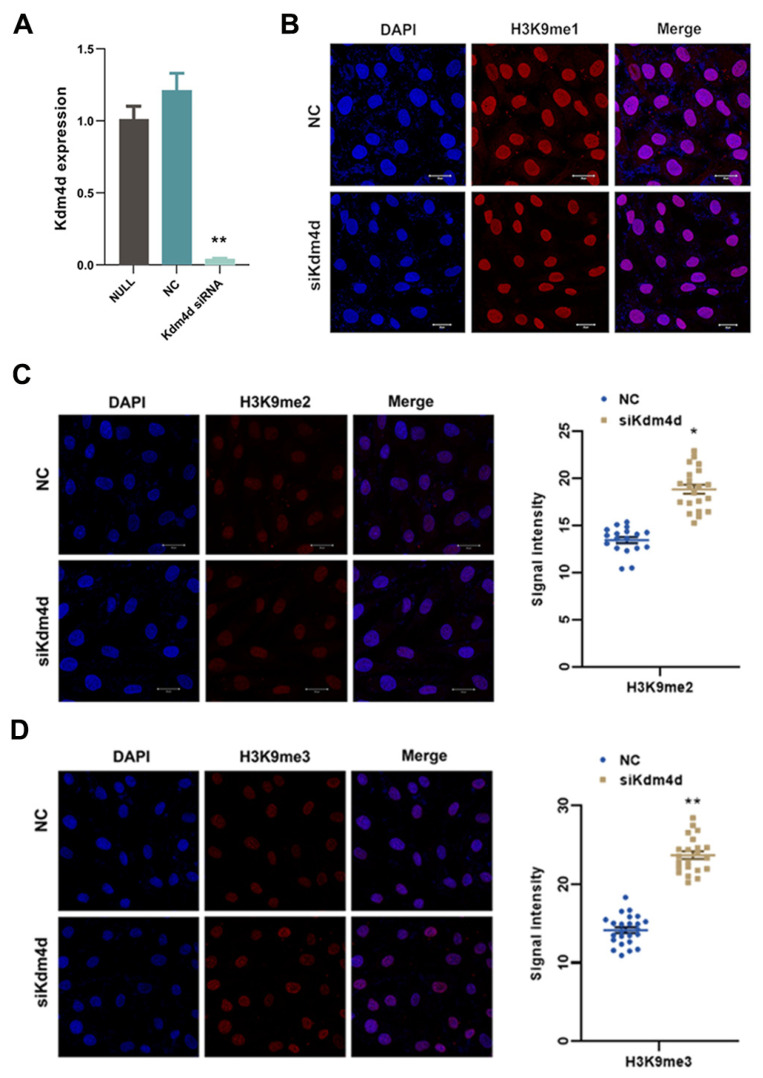
Increased H3K9 methylation in Kdm4d knockdown goat fibroblast cells. (**A**) The expression of Kdm4d was found to be significantly decreased in Kdm4d knocked-down goat fibroblast cells using qPCR; (**B**) the expression of H3K9me1 has no significant difference in Kdm4d knocked-down goat fibroblast cells, as revealed using cellular immunofluorescence staining; and (**C**,**D**) the expression of H3K9me2 and H3K9me3 was significantly decreased in Kdm4d knocked-down goat fibroblast cells. * denotes a significant difference at *p* < 0.05, ** denotes a significant difference at *p* < 0.01.

**Figure 3 biology-12-01095-f003:**
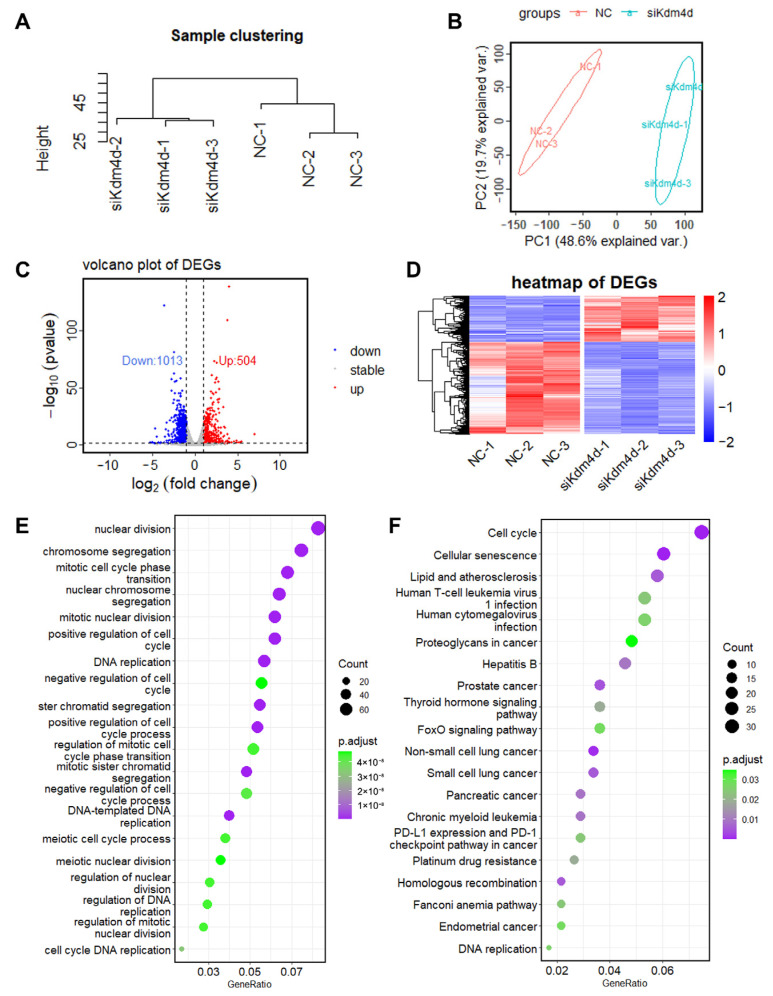
The transcription profile of Kdm4d knockdown goat fibroblast cells. (**A**) Sample cluster map; (**B**) principal component analysis (PCA); (**C**) volcano plot of the DEGs in normal goat fibroblast cells and Kdm4d knocked-down goat fibroblast cells; (**D**) heatmap of the differentially expressed RNAs in Kdm4d knockdown goat fibroblast cells; and (**E**,**F**) GO and KEGG enrichment analysis of the DEGs.

**Figure 4 biology-12-01095-f004:**
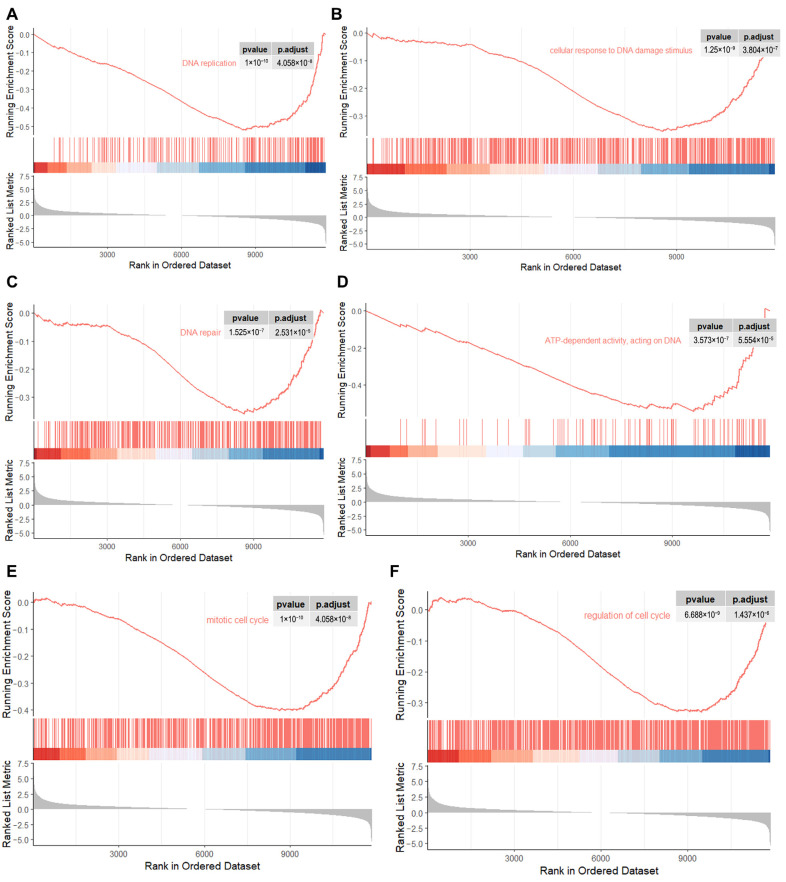
GSEA analysis revealing the critical pathways associated with the cell cycle. (**A**–**F**) The down-regulated genes were enriched in DNA replication, cellular response to the DNA damage stimulus, DNA repair, ATP-dependent activity, acting on DNA, the mitotic cell cycle, and in the regulation of the cell cycle pathway as revealed from GSEA analysis.

**Figure 5 biology-12-01095-f005:**
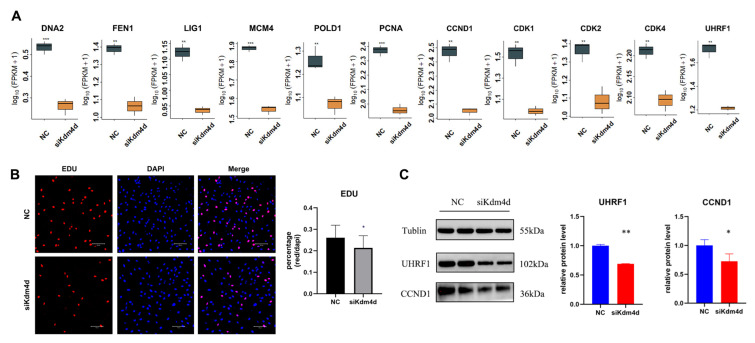
Kdm4d regulates the cell cycle of goat fibroblast cells. (**A**) The expression of genes associated with DNA replication and the cell cycle in Kdm4d knocked-down goat fibroblast cells; (**B**) cell proliferation in Kdm4d knocked-down goat fibroblast cells by EdU; (**C**) expression of cell cycle-related proteins in Kdm4d knocked-down goat fibroblast cells following Western blot analysis. * denotes a significant difference at *p* < 0.05, ** denotes a significant difference at *p* < 0.01, *** denotes a significant difference at *p* < 0.001.

**Figure 6 biology-12-01095-f006:**
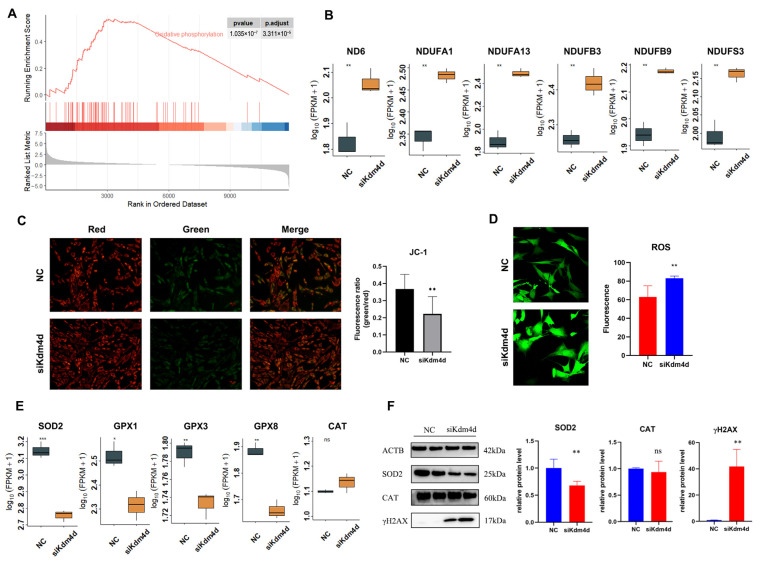
Kdm4d regulates the antioxidant capacity of goat fibroblast cells. (**A**) GSEA analysis enriched the oxidative phosphorylation pathway; (**B**) the expression of genes associated with oxidative phosphorylation in Kdm4d knocked-down goat fibroblast cells; (**C**) mitochondrial membrane potential changes in Kdm4d knocked-down goat fibroblast cells using JC-1; (**D**) real-time reactive oxygen levels in goat fibroblast cells after Kdm4d knockdown; (**E**) the expression of genes associated with antioxidation in Kdm4d knocked-down goat fibroblast cells; and (**F**) the expression of antioxidant-related proteins in Kdm4d knocked-down goat fibroblast cells. ns denotes a non-significant difference, * denotes a significant difference at *p* < 0.05, ** denotes a significant difference at *p* < 0.01, *** denotes a significant difference at *p* < 0.001.

**Table 1 biology-12-01095-t001:** Primer sequence information.

Item	Primer sequence (5′-3′)
Kdm4d	F: GGCAGATGTGGTTCTCGTCGTC
	R: GGTCTTGAGCCTTGCGGTCTAAG
ACTB	F: GGGAATCGTCCGTGACATCAA
	R: GTAGTTTCGTGAATGCCGCAG

**Table 2 biology-12-01095-t002:** siRNA sequence information.

Item	Sequence
siKdm4d	Sense: GAUAUUUCAUCCAACCAAATT
	Antisense: UUUGGUUGGAUGAAAUAUCAT

**Table 3 biology-12-01095-t003:** Protein antibody information.

Antibody	Cat No.	Source	Dilution of IF/WB
H3K9me1	Ab9045	Abcam(Cambridge, UK)	1:200
H3K9me2	4658T	CST(Danvers, MA, USA)	1:200
H3K9me3	Ab71604	Abcam(Cambridge, UK)	1:200
UHRF1	Ab194236	Abcam(Cambridge, UK)	1:1000
CCND1	60186-1-IG	Proteintech(CHI, USA)	1:1000
SOD2	24127-1-AP	Proteintech(CHI, USA)	1:1000
CAT	21260-1-AP	Proteintech(CHI, USA)	1:1000
H2AX	Ab124781	Abcam(Cambridge, UK)	1:200
Tublin	66031-1-lg	Proteintech(CHI, USA)	1:10,000
ACTB	Ab8227	Abcam(Cambridge, UK)	1:4000
Goat anti-rabbit IgG	31460	Pierce(Rockford, USA)	1:10,000
Goat anti-mouse IgG	SA00001-1	Proteintech(CHI, USA)	1:5000
Alexa Flour 488-conjugated anti-mouse IgG	ab150117	Abcam(Cambridge, UK)	1:200
Alexa Fluor 594 conjugate anti-rabbit IgG (H&L)	8889	CST(Danvers, MA, USA)	1:250

## Data Availability

The image and figure data used in this study have been included as attachments in this manuscript for reference. Molecular weight markers and other essential information have been labeled in the image data to ensure data accuracy and reproducibility. We encourage data sharing to promote transparency and collaboration in scientific research.

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
