# Peer review of "Lysine-Specific Demethylase 4D Is Critical for the Regulation of the Cell Cycle and Antioxidant Capacity in Goat Fibroblast Cells"

_biology, 2023, doi:10.3390/biology12081095_

Round 1

Reviewer 1 Report

In this manuscript, the authors showed that Kdm4d knockdown leads to decreased cell proliferation, characterized with decreased mRNA and protein expression of the cell cycle-related genes UHRF1 and CCND1, decreased antioxidant capacity, characterized with decreased mRNA and protein expression of SOD2, one of the major antioxidant enzymes, and increased DNA double-strand breaks, characterized with increased protein expression of γH2AX. Overall, the manuscript is well organized. Meanwhile, there are several concerns that may weaken the manuscript.

1. In the abstract, “In the present study, we identified 504 up-regulated and 1013 down-regulated genes 21 after knockdown of Kdm4d,” it’s better to specify knockdown of Kdm4d in what cells? Here I suppose it’s the goat fibroblast cells.

2. In the abstract, “suggesting vitol roles of Kdm4d in cell cycle and antioxidant regulation.” – vital?

3. causing DNA lesions – Please clarify what kind of lesions? DNA double strand breaks?

4. In the introduction section, the authors mention GPX4 in protection of oxidative stress-induced damage, however, in the results section, the authors showed the mRNA expression level of GPX1, 3, and 8, but not GPX4. Please clarify.

5. GFCs: I’m not sure what the abbreviation stands for. Is it goat fibroblast cells? Please provide the full name when this abbreviation first shows up in the manuscript. Also, is it derived from the goat testis? Please describe briefly how it’s generated. I understand the authors provided a reference. Still brief description will make the manuscript more readable. This is the same case for other methods in the methods section. Even though the authors provided a reference for each method, a brief description of the procedures is still needed.

6. The manuscript presented two phenotypes: proliferation and antioxidant capacity. Any reason that the authors just highlight one phenotype in the title?

Overall, the quality of English language is good.

Reviewer 2 Report

Manuscript ID: biology-2516297

Dear Authors,

I have received your manuscript entitled “Kdm4d is critical for regulation of antioxidant capacity in fibroblast cells” submitted to Biology, which describes the process of proving at the gene and protein level that kdm4d plays an important role in enhancing the antioxidant capacity of goat fibroblasts. The work is interesting and well-documented, but it has many editorial shortcomings.

The work is interesting and well-documented, but it has many editorial shortcomings.

Are following:

1. The title does not state that the work concerns goat fibroblasts.

2. There is not a word about them in the summary either.

3. The introduction does not contain the purpose of this work or its justification.

4. The materials and methods chapter is written nonchalantly as if everyone should know everything from previous publications (The work must be read with five cited previous works - it is not easy). The material is not described. Not all abbreviations are explained.

5. The results lack the title of Figure 2. There are no explanations for Figure 3D, on the basis of which parameter the heat map was constructed? (no markings, not explained). What cutoff values were used for the DEGs?

6. Conclusions are very general, as is the title. Do these conclusions apply to all fibroblasts, regardless of species? Goat or man?

English needs minor corrections, typos, and mistakes for no native.

Round 2

Reviewer 2 Report

Thank you. The work is now readable.

If your work exceeds the word limit, consider transferring the description of the methodology reproduced from your previous publications to supplementary materials and inform the reader of this.

Author Response

Dear Reviewer,
Thank you for reviewing our manuscript and providing valuable feedback. We appreciate the time and effort you have dedicated to evaluating our work.
After careful consideration, we have decided not to make any changes to the Materials and Methods section. We believe the current version adequately presents our research methods and experimental design, and complies with the requirement of having a minimum word count of 4000 words. We understand your concern regarding the length of the Materials and Methods section and have made efforts to ensure its clarity and conciseness without compromising essential information.
Once again, we express our gratitude for your insightful comments and suggestions. Your input has been instrumental in improving the quality of our manuscript.